# Combined Stimulation of the Substantia Nigra and the Subthalamic Nucleus for the Treatment of Refractory Gait Disturbances in Parkinson’s Disease: A Preliminary Study

**DOI:** 10.3390/jcm11082269

**Published:** 2022-04-18

**Authors:** Marta Villadóniga, Lidia Cabañes-Martínez, Laura López-Viñas, Samira Fanjul, Marta del Álamo, Ignacio Regidor

**Affiliations:** 1Department of Clinical Neurophysiology, Hospital Ramón y Cajal, 28034 Madrid, Spain; marta.villadoniga@salud.madrid.org (M.V.); lauralvinas@hotmail.com (L.L.-V.); ignacio.regidor@salud.madrid.org (I.R.); 2Department of Neurology, Hospital Ramón y Cajal, 28034 Madrid, Spain; samira.fanjul@salud.madrid.org; 3Department of Neurosurgery, Hospital Ramón y Cajal, 28034 Madrid, Spain; marta_delalamo@yahoo.es

**Keywords:** gait, deep brain stimulation, subthalamic nucleus, substantia nigra, Parkinson’s disease

## Abstract

Deep brain stimulation of the subthalamic nucleus is efficient for the treatment of motor symptoms (i.e., tremors) in patients with Parkinson’s disease. Gait disorders usually appear during advanced stages of idiopathic Parkinson’s disease in up to 80% of patients and have an important impact on their quality of life. The effects of deep brain stimulation of the subthalamic nucleus on gait and balance are still controversial. For this reason, alternative targets have been considered, such as stimulation of the pedunculopontine nucleus and the pars reticulata of substantia nigra, involved in the integration of the functional connections for gait. Due to the proximity of the subthalamic nucleus to the substantia nigra, their combined stimulation is feasible and may lead to better outcomes, improving axial symptoms. Our objective was to prospectively compare simultaneous stimulation of both structures versus conventional subthalamic stimulation in improving gait disorders. In ten patients with advanced Parkinson’s disease, deep brain stimulation leads (eight linear contacts) were implanted, and gait analysis was performed 6 months after surgery in off-stimulation and after 4 weeks of dual or single subthalamic stimulation. An improvement in gait parameters was confirmed with both stimulation conditions, with better results with combined substantia nigra and subthalamic stimulation compared with conventional subthalamic stimulation. Further studies are needed to determine if this effect remains after long-term dual-target stimulation.

## 1. Introduction

Deep brain stimulation of the subthalamic nucleus (STN-DBS) is an established surgical treatment for selected patients with Parkinson’s disease (PD) [1,2,3]. However, patients with prominent axial symptoms, i.e., gait disorders and instability, have not been considered good candidates, and the effects of STN-DBS on these symptoms remain controversial [4,5]. Gait disorders usually show up in advanced stages of idiopathic PD, appearing in up to 80% of patients [6,7]. This carries a detrimental impact on their quality of life [8]. Although there is evidence of the short-term beneficial effects of STN-DBS on gait and other axial symptoms [4,9], the long-term effect is a matter of debate [10,11]. For this reason, other lines of treatments have been considered, such as different stimulation parameters [12,13], stimulation of the Pedunculopontine nucleus (PPN) [14] and the pars reticulata of Substantia nigra (SNr) [4,15,16] involved in the integration of the functional connections for gait [17]. Due to the proximity of the STN to the SNr, their simultaneous stimulation is feasible and may lead to better outcomes and improved gait results [16,18].

The objective of our study was to demonstrate the beneficial effects of dual STN + SNr stimulation on gait and to test the hypothesis that STN + SNr-DBS is superior to STN-DBS in improving gait disturbances in advanced PD patients.

## 2. Materials and Methods

### 2.1. Design

This is a prospective, randomized, cross-over-2×2, double-blinded study carried out in a tertiary hospital in Madrid. Six months after surgery, we compared the effects on gait (through quantitative analysis of the percentage of complete cycles with a normal gait sequence) of dual stimulation (STN + SNr-DBS) with single stimulation (STN-DBS) after 4 weeks of treatment with each type of stimulation.

### 2.2. Subjects

Ten patients (6 male, 4 female, age 56.8 ± 5.6 years) suffering from advanced idiopathic PD (disease duration 6–15 years) participated in the study. They were treated with implantation of a linear octopolar electrode for DBS (Boston Scientific, Natick, MA, USA) and a rechargeable implantable pulse generator (Vercise™, Boston Scientific, Natick, MA, USA). We included patients with ages between 18 and 70 years, diagnosed with idiopathic PD (according to the British Brain Bank criteria) at an advanced stage, with a good response to levodopa but with refractory fluctuations and medication-induced dyskinesia, and with a disease duration of more than 5 years. All patients were candidates for DBS of the STN and SNr and had clinical, radiological and electrophysiological testing that verified the proper placement of electrodes in STN and SNr. Dopaminergic medication intake was unchanged for at least 4 weeks before surgical intervention. All patients signed informed consent before intervention. Exclusion criteria: pregnancy or planning to become pregnant during the course of the study, cognitive impairment (Mini-Mental State Exam under 25), participation in another study during the clinical trial or 3 months before, serious psychiatric or other severe pathological condition and acute adverse events arising from STN + SNr-DBS.

### 2.3. Study Protocol

Upon enrollment, the following evaluations were carried out (Figure 1):Baseline: At least 6 months after surgery, the period during which the patients were receiving STN-DBS, quantitative gait analysis was performed. All patients were in off-stimulation (DBS stopped at least 48h before the analysis) and off-medication state (having stopped taking the medication for a minimum of 12 h before). After baseline evaluation, patients were randomized to be programmed in the corresponding stimulation mode (STN-DBS versus STN + SNr-DBS);Visit 1: At the end of the first period, after four weeks, quantitative gait analysis in the off-medication state was repeated. Subsequently, the stimulation type was changed (STN + SNr-DBS vs. STN-DBS) for each group (cross-over);Visit 2: After the second period, lasting 4 weeks, quantitative gait analysis in the off-medication state was repeated.

### 2.4. Locating the Electrodes

The spatial location of contacts on STN and SNr was determined by image fusion of the magnetic resonance (MR) image taken prior to surgery with the postoperative CT scan using a software System. Implantation of the octopolar electrode for DBS (Boston Scientific, Natick, MA, USA) was performed under local anesthesia using a Leksell (Elekta, Stockholm, Sweden) stereotactic frame at the 3D visualization (x, y and z coordinate system) for the location of STN and SNr. The intervention was performed in an off-medication state, and intraoperative single-cell recordings were used to confirm STN and SNr locations and determine the optimal electrode placement. The final location of the electrode was confirmed by neuronal activity and by relief of symptoms without adverse effects. The implantable pulse generator (VerciseTM, Boston Scientific, Natick, MA, USA) was placed in the subclavicular area and connected subcutaneously to the intracerebral electrodes. The electrodes and their anatomical positioning were visualized using 3D reconstruction software Guide XT (Boston Scientific Corp., Boston, MA, USA), verifying their position in STN and SNr.

### 2.5. Stimulation Parameters

New electric implantable pulse generators allowed us to distribute electric pulses using multiple independent current control (MICC) to fractionalize current so that the stimulation pulses were distributed at the same time between different contacts. Thus, each contact could be programmed with a specific amplitude while keeping the same stimulation frequency and pulse width for all contacts within one area, therefore enabling optimization of stimulation parameters, minimizing adverse effects [19]. The use of MICC allowed simultaneous stimulation with specific parameters in both areas, STN and SNr (Figure 2). With the selected stimulation parameters, we were able to stimulate both nuclei to maximize the patient’s clinical benefits. Motor symptoms were optimally controlled with the standard stimulation of the STN. The stimulation for both nuclei was delivered using a pulse duration of 60 µs and a stimulation frequency of 130 Hz.

### 2.6. Gait Analysis

Quantitative gait analysis was carried out using a multichannel recording device STEP 32 (DemItalia, Volpago del Montello, Italy). STEP 32 is a computerized system of movement analysis, which automatically records podobasographical signals and analyzes step cycle characteristics. In addition, this system allows simultaneous video camera recording. For the acquisition of the podobasographical signals, 6 foot switches were placed on the sole of both feet, distributed in the heel bone and the heads of the first and fifth metatarsi (Figure 3). The following parameters were obtained: the number of total cycles and percentage of cycles with normal sequence (heel strike, foot contact, push off and swing; HFPS). Participants were required to walk a distance of around 40 m. In order to obtain reliable results, we studied gait cycles relative to strides recorded during a walk along a straight walkaway; subjects were asked to walk back and forth the corridor twice. Strides involving deceleration, reversing and acceleration were removed automatically by the system, using a multivariate statistical filter that detected and eliminated outliers [20]. Given that currently, they are no normative values regarding the normal percentage of normal cycles, we considered the published data, and assumed that in a healthy subject, the percentage of atypical cycles is around 10% [21,22]. Accordingly, in subjects with gait impairment, the percentage of atypical cycles would be increased, and this fact may be used as a reliable outcome measure to evaluate even subtle improvements [20,21].

### 2.7. Statistical Analysis

For the purpose of evaluating the effectiveness of dual-target stimulation, assuming an improvement of 20% in the quantitative analysis of the STEP32 gait, a paired t-test was used. For statistical analysis, quantitative gait data from STEP32 were used to compare the effects of 4 weeks of stimulation of the STN versus dual-target stimulation of STN + SNr. The mean and standard error of the mean (SEM) were used due to their normal distribution. We used a paired t-test to test the null hypothesis of equality of these two conditions. Analysis was performed by means of the intent-to-treat strategy. The level of statistical significance required was set to *p* < 0.05, with bilateral contrast. All statistical analyses were carried out using Stata software (version 13.1. StataCorp. 2013. Stata Statistical Software: Release 13., StataCorp LP, College Station, TX, USA).

## 3. Results

A difference in the percentage of effective steps in the neurophysiological analysis of gait was found. There is a statistically significant difference in the mean of the normal cycles between STN-DBS and STN + SNr-DBS (*p* = 0.038). Data are summarized in Table 1 and Figure 4.

## 4. Discussion

In this randomized, double-blind, cross-over preliminary study, the effect of combined STN + SNr DBS compared to conventional STN-DBS on gait was studied by a quantitative analysis of gait using the STEP32 system, with the aim of determining the beneficial effects of combined stimulation (STN + SNr-DBS) on gait, and to test the hypothesis that STN + SNr-DBS is superior to STN-DBS for improving gait disturbances on advanced PD patients.

Previous reports described that the frequency of correct sequence cycles (HFPS) is markedly reduced in PD patients [21,23], even in the on-medication state. This parameter could be sensitive to detecting gait instability [20]. In previous studies by our group, this parameter improved markedly in PD patients after STN-DBS, along with the clinical scales (data not published). Here we investigated the percentage of cycles with normal sequence (HFPS) using quantitative gait analysis performed by STEP 32 after four weeks of constant stimulation in either (STN-DBS or STN + SNr-DBS) conditions. After the first 4 weeks, we changed to the other type of stimulation, and the patient remained with this new stimulation for another period of 4 weeks. The results of our study show that there is a significant difference between the pre-and post-surgical situation with regards to the percentage of complete cycles with a normal gait sequence (regardless of the stimulation type: STN-DBS, documenting previous observations in our clinic, or combined STN + SNr-DBS). Furthermore, there was a statistically significant difference in the improvement between these two types of stimulation, with better results when using the combined STN + SNr-DBS paradigm. In our opinion, this finding is especially relevant since spatial-temporal parameters of gait, as the percentage of complete gait cycles, reflect indirect biomarkers for the clinical phenomenon of freezing of gait [24], and their modulation through DBS plays an important role in the clinical improvement of PD patients.

The beneficial effects of simultaneous STN-SNr-DBS on gait [16,18,24,25,26] and other disabling parkinsonian symptoms [27,28] were already described. Our results are in accordance with previous reports, revealing a favorable effect of combined STN + SNr–DBS on gait in PD patients, and confirm the results of the only previous series of cases describing the effects on gait of combined DBS through quantitative gait analysis [24]. STEP 32 gait system equipment allows us to measure an important spatial gait characteristic, the percentage of steps with normal sequence. This parameter is reduced in patients with other pathologies with prominent gait disturbances [21,29] and with idiopathic PD [23] and could be sensitive for detecting gait instability. In our study, this parameter not only showed a statistically significant improvement after STN-DBS, probably reflecting an improvement in the patients’ stability, but the betterment was even more evident after STN + SNr-DBS.

The effects of low and high stimulation frequencies on gait and other axial symptoms were described [30,31]. Valldeoriola et al. described the beneficial effect of combined simultaneous high-frequency STN-DBS with low-frequency SNr-DBS [18]. Despite their promising results, we decided to perform both stimulations with high frequency, as described in previous reports supporting the use of this type of stimulation [16,31,32], based on the hypothesis that high-frequency inhibitory stimulation of the SNr releases the excessive basal ganglia inhibitory tone on the brainstem regions involved in the control of locomotion and posture [24,33,34]. Our results reveal a beneficial effect of STN + SNr-DBS on spatial gait characteristics compared to STN-DBS alone, supporting this hypothesis.

We are aware of the limitations of our study. The small sample size of our series of patients, together with the heterogeneous presentation of gait disturbances in PD patients, may bias our results, and thus our results should be considered preliminary. Further studies enrolling more patients and even multicenter studies would be necessary to draw stronger conclusions.

## 5. Conclusions

In summary, our work supports the hypothesis that simultaneous stimulation of the STN and SNr improves the short-term effects on gait disturbances, proving better outcomes than conventional stimulation of the STN-DBS. It is necessary to develop further investigations to study the long–term effects of this combined STN + SNr-DBS.

## Figures and Tables

**Figure 1 jcm-11-02269-f001:**
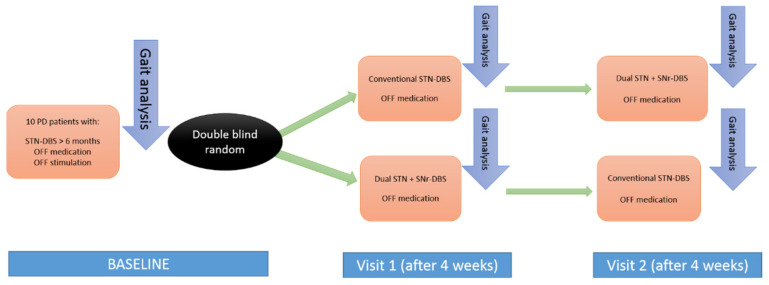
Representation of the protocol via block diagram. PD: Parkinson’s disease. STN-DBS: Subthalamic deep brain stimulation. STN + SNr-DBS: Dual subthalamic and nigral deep brain stimulation.

**Figure 2 jcm-11-02269-f002:**
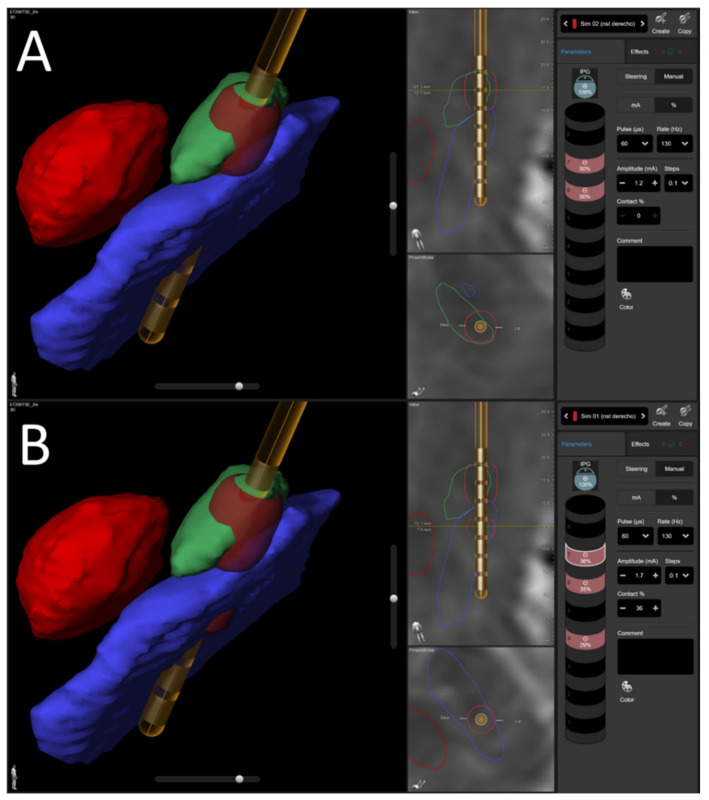
Stimulation Field Model of the STN (**A**) and STN+SNr (**B**). Green = STN; Blue = SNr; Ferrari Red = Red Nucleus; Dusty Red = Volume Tissue Activated.

**Figure 3 jcm-11-02269-f003:**
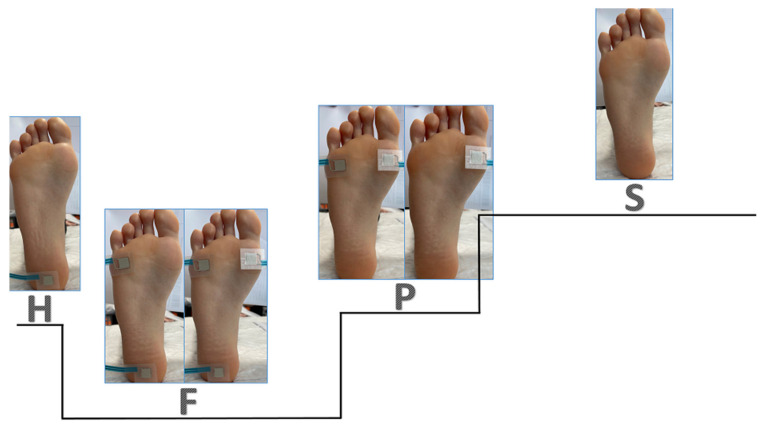
Gait cycle. H: heel contact, F: foot contact, P: push off, S: swing.

**Figure 4 jcm-11-02269-f004:**
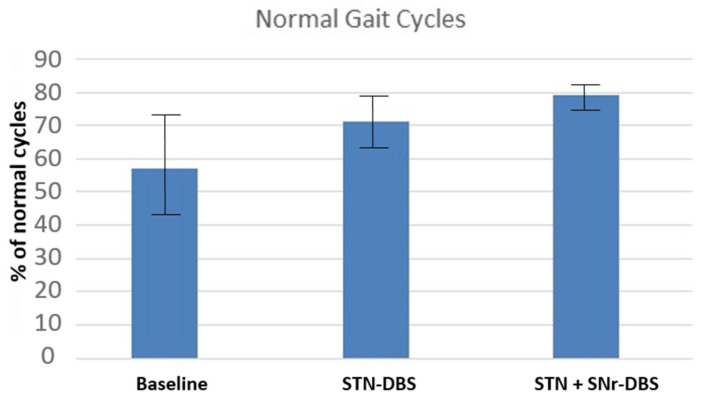
Differences of mean between baseline and the different STN parameters in percentage of cycles with the right sequence (HFPS). STN: Subthalamic nucleus stimulation; STN + SNR: Dual Subthalamic and nigral stimulation.

**Table 1 jcm-11-02269-t001:** Summary of gait analysis data HFPS: heel strike, foot contact, push off and swing; STN: Subthalamic nucleus stimulation; STN + SNR: Dual Subthalamic and nigral stimulation; SEM: Standard error of the mean.

HFPS Cycles	Baseline	STN	STN + SNr
Mean (± SEM)	0.57 (0.41; 0.73)	0.71 (0.63; 0.79)	0.79 (0.74; 0.84)

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
