# Peer review of "Combined Stimulation of the Substantia Nigra and the Subthalamic Nucleus for the Treatment of Refractory Gait Disturbances in Parkinson’s Disease: A Preliminary Study"

_jcm, 2022, doi:10.3390/jcm11082269_

Round 1
Reviewer 1 Report
This is a study presenting the effects of combined stimulation of the the substantia nigra and the subthalamic nucleus in the treatment of gait disturbances. There are certain concerns that should be addressed before further evaluation of the manuscript:
- Authors state that: "Gait disorders usually show up in advanced stages of idiopathic PD, 34 apearing in up to 80% of patients", which is true, however due to the small number of patients in the study authors should take into account possible limitations as overlapping clinical manifestations of PD and PSP-P (possible misdiagnosis regardless of using the criteria of diagnosis) when attempting to evaluate larger groups of patients in the the future. (References: >>> Progressive Supranuclear Palsy-Parkinsonism Predominant (PSP-P)-A Clinical Challenge at the Boundaries of PSP and Parkinson's Disease (PD). Front Neurol. 2020 Mar 10;11:180. doi: 10.3389/fneur.2020.00180. PMID: 32218768; PMCID: PMC7078665. >>> "Parkinson's disease" on the way to progressive supranuclear palsy: a review on PSP-parkinsonism. Neurol Sci. 2021 Dec;42(12):4927-4936. doi: 10.1007/s10072-021-05601-8. Epub 2021 Sep 17. PMID: 34532773.
- Regarding the small number of the small number of patients accompanied by diverse disease duration, the preliminary character of the study should be stressed in the title as well as throughout the manuscript.
- An extended desription of the current neurological status of the included patients would be beneficial
- More data concerning the pharmacological treatment of the patients should be added - response to levodopa treatment.
- A separate paragraph highlighting the limitations should provide a more extended overview.
Author Response
This is a study presenting the effects of combined stimulation of the substantia nigra and the subthalamic nucleus in the treatment of gait disturbances. There are certain concerns that should be addressed before further evaluation of the manuscript: Authors state that: "Gait disorders usually show up in advanced stages of idiopathic PD, 34 apearing in up to 80% of patients", which is true, however due to the small number of patients in the study authors should take into account possible limitations as overlapping clinical manifestations of PD and PSP-P (possible misdiagnosis regardless of using the criteria of diagnosis) when attempting to evaluate larger groups of patients in the the future. (References: >>> Progressive Supranuclear Palsy-Parkinsonism Predominant (PSP-P)-A Clinical Challenge at the Boundaries of PSP and Parkinson's Disease (PD). Front Neurol. 2020 Mar 10;11:180. doi: 10.3389/fneur.2020.00180. PMID: 32218768; PMCID: PMC7078665. >>> "Parkinson's disease" on the way to progressive supranuclear palsy: a review on PSP-parkinsonism. Neurol Sci. 2021 Dec;42(12):4927-4936. doi: 10.1007/s10072-021-05601-8. Epub 2021 Sep 17. PMID: 34532773. We agree with this point. In our series of patient, given the advanced state of the disease, and without any symptoms suggesting a different diagnosis, as PSP, the misdiagnosis was unlikely. Nevertheless, if we try to evaluate larger populations of patients in the future, and especially if we treat patients in early stages, we must be very careful not to include PSP-P patients, as DBS would not be an adequate choice of treatment. Regarding the small number of the small number of patients accompanied by diverse disease duration, the preliminary character of the study should be stressed in the title as well as throughout the manuscript. We agree with this point and we have stressed it in the title as well as in the discussion. An extended description of the current neurological status of the included patients would be beneficial A comprehensive description of the current neurological status of the patients is the object of an ongoing follow up paper, which we are currently preparing including the clinical and neurophysiological data. More data concerning the pharmacological treatment of the patients should be added - response to levodopa treatment. One of the criteria to select patients for DBS was the good response to levodopa. This information has been added to the manuscript. A separate paragraph highlighting the limitations should provide a more extended overview. The limitations paragraph has been somehow changed.Reviewer 2 Report
- Gait analysis paragraph, please include normative values for the parameters of gait studied if they are available for healthy subjects. If not then in the Gait analysis paragraph include an explanation for baseline measurement. Maybe to include some photo of placement of 6 foot switches. Maybe also to include in more detail technical procedures for analyzing cycle characteristics.
- The Table in the results needs to be properly positioned/cited in the manuscript text and given the title. For± SEM please explain two values in the brackets
- Figure 2 is not cited in the manuscript text. The resolution of Fig 2 is not good.
- Figure 3 needs to be better presented/cited in the text, not just “Figure 3”.
- Figure 3 missing the measuring unit on y-axis
- Please include more clearly in the Results section the obtained results for the studied parameters: number of total cycles and percentage of cycles with normal sequence for each subject (text and Table) for STN-DBS and STN+SNr-DBS.
- Figure 3 lacks the indication of a significant difference between the conditions
Author Response
- Gait analysis paragraph, please include normative values for the parameters of gait studied if they are available for healthy subjects. If not then in the Gait analysis paragraph include an explanation for baseline measurement. Maybe to include some photo of placement of 6 foot switches. Maybe also to include in more detail technical procedures for analyzing cycle characteristics.
More information regarding these concerns has been added to the manuscript. Also, a figure has been added (figure 3).
- The Table in the results needs to be properly positioned/cited in the manuscript text and given the title. For± SEM please explain two values in the brackets:
Agreed, the table has been repositioned and cited. The values in the brackets have been explained.
- Figure 2 is not cited in the manuscript text. The resolution of Fig 2 is not good.
Figure 2 is indeed cited in the manuscript, within the “stimulation parameters” paragraph. The resolution of the figure has been improved.
- Figure 3 needs to be better presented/cited in the text, not just “Figure 3”.
Figure 3 (now figure 4) has been better presented in the text.
- Figure 3 missing the measuring unit on y-axis
- Figure 3 lacks the indication of a significant difference between the conditions
Figure 3 (now figure 4) has been changed accordingly.
- Please include more clearly in the Results section the obtained results for the studied parameters: number of total cycles and percentage of cycles with normal sequence for each subject (text and Table) for STN-DBS and STN+SNr-DBS.
Given the limited number of patients of our study, we believe that including the individual data will not add much to the manuscript, as we are trying to demonstrate the preliminary global result of this therapeutic strategy. Regardless, if you consider including this data a fundamental factor for the acceptation of the manuscript, we are willing to send them for your evaluation
Round 2
Reviewer 1 Report
Authors have implemented some of my suggestions, however the introduction as I mentioned in my previous review, should be changed. Though the work is considering PD, in my opinion regarding the fact that the study is based on small number of patients and wider examinations are affected by the risk of including PSP-P patients, the introduction should also indicate that:
"Gait disorders usually show up in advanced stages of idiopathic PD, apear- 34 ing in up to 80% of patients" however these elements of clinical manifestation can be also observed in PSP-P. Regarding the latest works (as mentioned in the first stage of review) on PSP-P, a short elaboration in the introduction would be beneficial.
Author Response
With regards to the PSP-P comment, we have changed the introduction accordingly.